# MAMBA NEURAL OPERATOR: WHO WINS? TRANSFORMERS VS. STATE-SPACE MODELS FOR PDES

## ABSTRACT

Partial differential equations (PDEs) are widely used to model complex physical systems, but solving them efficiently remains a significant challenge. Recently, Transformers have emerged as the preferred architecture for PDEs due to their ability to capture intricate dependencies. However, they struggle with representing continuous dynamics and long-range interactions. To overcome these limitations, we introduce the Mamba Neural Operator (MNO), a novel framework that enhances neural operator-based techniques for solving PDEs. MNO establishes a formal theoretical connection between structured state-space models (SSMs) and neural operators, offering a unified structure that can adapt to diverse architectures, including Transformer-based models. By leveraging the structured design of SSMs, MNO captures long-range dependencies and continuous dynamics more effectively than traditional Transformers. Through extensive analysis, we show that MNO significantly boosts the expressive power and accuracy of neural operators, making it not just a complement but a superior framework for PDE-related tasks, bridging the gap between efficient representation and accurate solution approximation.

## 1 INTRODUCTION

Partial differential equations (PDEs) describe various real-world phenomena, such as heat transfer (Heat Equation), fluid dynamics (Navier-Stokes), and biological systems (Reaction-Diffusion). While analytical solutions are sought, many PDEs—like the Navier-Stokes equations—lack closed-form solutions, making them computationally intensive to solve. Numerical methods, such as finite element, finite difference (Mehra et al., 2010), and spectral methods, discretise these equations but involve trade-offs between computational cost and accuracy. Coarser grids reduce computational load but sacrifice precision, while finer grids increase both accuracy and computational expense. Recent advancements in deep learning have changed techniques for solving PDEs. Physics-Informed Neural Networks (PINNs) (Raissi et al., 2019; Mattey and Ghosh, 2021) integrate governing equations and boundary conditions into the loss function, but often struggle with generalisation and require retraining for changes in coefficients. Neural operators (Bhattacharya et al., 2021; Kovachki et al., 2023), on the other hand, learn mappings between function spaces, offering a mesh-free, data-driven approach that generalises better across different PDE instances.

Operator learning has gained traction with models like DeepONet (Lu et al., 2019) and the Fourier Neural Operator (FNO) (Li et al., 2020a), which achieved state-of-the-art performance. These models learn input-output mappings to approximate complex operators, similar to sequence-to-sequence problems. Transformers (Vaswani, 2017) have become a go-to architecture for PDEs (Cao, 2021; Li et al., 2022a; Bryutkin et al., 2024) due to their ability to capture long-range dependencies. However, their quadratic complexity limits efficiency for tasks such as long-time integration. To overcome this, efficient variants like Galerkin attention (Cao, 2021) reduce computational cost to linear scaling. While these models improve efficiency, they trade off model capacity by approximating the self-attention mechanism, potentially reducing accuracy for tasks that need precise attention. Moreover, Transformers face challenges with PDEs due to limited context windows, inefficiency with continuous data, and high memory usage, making them less effective for capturing dependencies over continuous domains and high-resolution grids.

While Transformers are popular for PDE modelling, they have limitations in handling continuous data and high-resolution grids. An emerging alternative is State-Space Models (SSMs) (Gu et al., 2021; 2022), which offer better scalability, reduced memory usage, and improved handling of long-range dependencies in continuous domains compared to Transformers. In particular, Mamba (Gu and Dao, 2023) is a novel way designed to effectively capture long-range dependencies, handle continuous data efficiently, and reduce memory consumption in sequence-to-sequence problems. Although Transformers dominate applications like foundational models and computer vision, *the use of SSMs—especially Mamba—for neural operators in PDEs remains underexplored, and their theoretical connections and potential advantages are yet to be fully understood.*

**Contributions.** We introduce the concept of Mamba Neural Operator (MNO), which provides a novel perspective applicable to Transformer-based techniques for PDEs. Unlike closely related works, we offer a formal theoretical connection between Mamba and Neural Operators, demonstrating its advantages for PDEs. MNO addresses key challenges in PDE modelling by leveraging its structured state-space design to capture long-range dependencies and continuous dynamics more effectively than Transformers. Our particular contributions are as follows.

★ We introduce the concept of the Mamba Neural Operator (MNO), where we underline:

- Mamba Neural Operator expands the SSM framework into a unified neural operator approach, making it adaptable to diverse architectures, including any Transformer-based model.
- Unlike existing related works, we provide a theoretical understanding that shows how neural operator layers share a comparable structural framework with time-varying SSMs, offering a new perspective on their underlying principles.

★ We evaluate MNO on various architectures and PDEs, showing through systematic analysis that Mamba enhances the expressive power and accuracy of neural operators. This indicates that Mamba is not just a complement to Transformers, but a superior framework for PDE-related tasks, bridging the gap between efficient representation and accurate solutions.

## 2 RELATED WORK

● **Data-Driven PDEs.** Recent advances in fluid dynamics and solving PDEs have led to architectures modelling continuous-time solutions and multiparticle dynamics (Kochkov et al., 2021; Lusch et al., 2018). Physics-informed models now offer solutions in unsupervised and semi-supervised settings (Raissi et al., 2019; Li et al., 2020a). These models typically encode spatial data and evolve over time, utilising methods like convolutional layers (Ronneberger et al., 2015; Wiewel et al., 2019) symbolic neural networks (Udrescu and Tegmark, 2020), and residual networks (He et al., 2016). Finite element methods (FEM), including Galerkin and Ritz, are also integrated into learning frameworks (Chen et al., 2021).

● **Neural Operators.** Neural operators, such as the Graph Neural Operator (Li et al., 2020b) and Fourier Neural Operator (Li et al., 2020a), excel at learning mappings in infinite-dimensional spaces, particularly by leveraging techniques like graph structures or transformations in Fourier space. The Fourier Neural Operator (FNO) and its variations, including the incremental, factorised, adaptive FNO, and FNO+ (Zhao et al., 2022; Tran et al., 2021; Guibas et al., 2021) have shown exceptional performance in both speed and accuracy. Their key advantage lies in their ability to maintain discretisation invariance, which sets them apart in many applications. DeepONet (Lu et al., 2019) pioneered the nonlinear operator approximation using separate networks for inputs and query points, while extensions like MIONet handle multiple inputs (Jin et al., 2022). Challenges like irregular grids are being addressed through grid mapping and subdomain partitioning (Li et al., 2022b; Wen et al., 2022) though scalability for diverse inputs remains a key focus.

● **Transfomers for PDEs.** The Transformer model (Vaswani, 2017) stands out due to its distinctive features, primarily its use of attention mechanisms to model the relationships among input elements. Initially, it was developed for NLP, and attention mechanisms have been adapted to PDEs, providing flexible and efficient mappings between function spaces. Galerkin attention (Cao, 2021) introduced linear complexity to reduce computational costs, inspiring further developments like GNOT (Hao et al., 2023) and OFormer (Li et al., 2022a), which achieve state-of-the-art results. Additionally,

graph-based Transformers have also been explored to capture complex interactions in irregular domains (Bryutkin et al., 2024).

● **State-Space Models for PDEs & Comparison to Ours.** Initial studies on SSMs for PDEs, like MemNO (Buitrago Ruiz et al., 2024), explored combining FNO with S4 but were restricted to low-resolution or noisy inputs. In contrast, we introduce the Mamba Neural Operator, which generalises the SSM framework to neural operators, making it compatible with any architecture, including Transformers. Our approach extends the theoretical foundations for broad applicability to any PDE family, highlighting Mamba's effectiveness in diverse scenarios. At the time of our submission, the work of that (Hu et al., 2024) proposed integrating state-space models into neural operators for dynamical systems. While related, our work differs significantly, their approach focuses on dynamical systems and tests only on ordinary differential equations (ODEs), whereas we target parametric partial differential equations (PDEs). Additionally, we provide a theoretical understanding showing that neural operator layers share a comparable structural framework with time-varying SSMs, demonstrating alignment between hidden space updates and the iterative process in neural operators.

## 3 MAMBA NEURAL OPERATOR

This section details the theoretical underpinning and practicalities of the Mamba Neural Operator. We outline its design, key components, and operational mechanisms, explaining how it efficiently models partial differential equations by leveraging structured state-space models (SSMs).

### 3.1 PROBLEM STATEMENT

We consider parametric partial differential equations (PDEs) defined on a domain $\Omega \subset \mathbb{R}^n$, parameterised by $\theta \in S \subset \mathbb{R}^p$, where $\theta$ is sampled from a distribution $w$. The general form of the PDE is:

$$P : \mathcal{P} \times \Omega \times \mathcal{W} \times \mathbb{R}^m \times \ldots \times \mathbb{R}^m \to \mathbb{R}^\ell, \quad \Omega \subset \mathbb{R}^n, W \subset \mathbb{R}^m,$$
$$P(\theta, x, u, \partial_{x_1} u, \ldots, \partial_{x_n} u, \ldots, \partial_{x_1}^{\beta_1} \cdots \partial_{x_n}^{\beta_n} u) = 0,$$
(1)

where the unknown function $u : \Omega \to V$ solves $P$. The multi-index $\beta = (\beta_1, \ldots, \beta_n)$, with $|\beta| = \sum_{i=1}^n \beta_i$, determines the differentiation orders. If time is involved, $\Omega$ reduces to $\mathcal{T} \subset \mathbb{R}_{\geq 0}$ and $\Omega \subset \mathbb{R}^{n-1}$. To ensure well-posedness, initial and boundary conditions must hold:

$$u(x, T_0) = u_0(x), \ x \in \Omega_\theta, \qquad u(x, t) = u_b(x), \ x \in \partial\Omega_\theta, \ t \in \mathcal{T},$$
(2)

for $x \in \Omega_\theta$ and $t \in \mathcal{T}$, where $u_0$ and $u_b$ are the initial and boundary conditions, respectively. Assume $\Omega, \mathcal{P}, V$ are Banach spaces, and there exists an analytic solution operator: $O : \mathcal{P} \times \Omega \times \mathbb{R}^m \times \ldots \times \mathbb{R}^m \times \mathbb{R}^\ell \times \mathbb{R}^\ell \to V$. Our aim is to design a neural network $\tilde{S}\mu : (\theta, u_0, u_b) \mapsto u$ that approximates this operator, with $\mu$ as the network's parameters. Given a dataset $(\theta^{(n)}, u^{(n)})n = 1^N$, where $\theta^{(n)}$ and $u^{(n)}$ correspond to the system's discretised parameters, we simplify the notation as $\theta^{(n)} = \theta(x^{(n)})$ and $u^{(n)} = u(x^{(n)})$.

### 3.2 PRELIMINARIES: MAMBA

Transformers are the leading architecture for many state-of-the-art techniques for PDEs, with preliminaries introduced in Appendix A. In this section, we outline the background on State Space Sequence models (SSM). Structured State Space (S4) models introduce a new approach in deep learning sequence modelling, incorporating elements from Recurrent Neural Networks (RNNs), Convolutional Neural Networks (CNNs), and classical state space models. These models are inspired by control theory, where the process involves mapping an input sequence $u(t) \in \mathbb{R}^L$ to an output sequence $y(t) \in \mathbb{R}^L$ through a hidden latent state $h(t) \in \mathbb{R}^N$. The core mechanism of State Space Models (SSMs) is formulated using linear first-order ordinary differential equations, enabling efficient handling of temporal data, which reads:

$$h'(t) = Ah(t) + Bu(t), \quad y(t) = Ch(t) + Du(t),$$
(3)

where $A \in \mathbb{C}^{N \times N}$ and $B, C \in \mathbb{C}^N$. Mamba, a more advanced variant of SSMs, refines this formulation by incorporating efficient state space parameterisation and selection mechanisms. Unlike

earlier models such as S4, which uses bilinear method, Mamba adopts zero-order holds, allowing it to handle larger hidden states and longer sequences more effectively. This makes Mamba particularly well-suited for complex sequence modeling tasks, such as natural language processing and time-series analysis.

### 3.3 State Space Models Discretisation for PDEs

State Space Models (SSMs) have emerged as a strong alternative to Transformers in deep learning. While Transformers dominate in areas like foundational models and computer vision, the application of SSMs, particularly the Mamba architecture, to neural operators for PDEs is still underexplored.

We start by demonstrating that the discretisation of S6 (Mamba) is equivalent to the well-known Eular method when the Taylor series expansion is applied. Mamba utilises zero-order holds, resulting in the following discretisation, which reads:

$$A = \exp(\Delta A), \quad B = (\Delta A)^{-1} (\exp(\Delta A) - I) \cdot \Delta B. \tag{4}$$

**Discretisation of SSM.** To incorporate the SSM into deep learning frameworks, we need to transform the continuous-time SSM into a discrete formulation. This is done by expressing the continuous-time system as an ordinary differential equation (ODE) and then solving it numerically. As discussed in (Liu et al., 2024), the discrete SSM reads:

$$
\begin{aligned}
h_{a+1} &= e^{A\Delta a}(h_a + B_a U_a e^{-A\Delta a \Delta a}) \\
&= e^{A\Delta a} h_a + B_a \Delta_a u_a = \bar{A}_a h_a + \bar{B}_a u_a,
\end{aligned} \tag{5}
$$

where $\Delta$ is the time step size, and $\bar{A}_a = e^{A\Delta a}$ and $\bar{B} = B_a \Delta_a$ are the discretised system matrices. In the S6 model, we define $\tilde{A} = e^{\Delta A}$ and $\tilde{B} = (\Delta A)^{-1}(e^{\Delta A} - I) \cdot \Delta B$. By applying sampling in $\tilde{A}$, we have $\bar{A} = \tilde{A}$. For $\tilde{B}$, applying the sampling process yields to:

$$
\begin{aligned}
\tilde{B} &= (\Delta A)^{-1}(e^{\Delta A} - I) \cdot \Delta B = (\Delta A)^{-1}(I + \Delta A + O(\Delta^2) - I) \cdot \Delta B \\
&= (\Delta A)^{-1}(\Delta A + O(\Delta^2)) \cdot \Delta B = \Delta B(Drop \, O(\Delta^2)) = \bar{B}
\end{aligned} \tag{6}
$$

Thus, we have shown that our discretisation method is equivalent to the zero-order hold method, where $\bar{A} = \tilde{A}$ and $\bar{B} = \tilde{B}$.

**Proposition 1.** *The zero-order hold discretisation method, as in (4), is equivalent to the Euler method in SSM when the Taylor series expansion of the exponential function is truncated to its first-order term.*

*Proof.* In SSM, if we define the matrices as $\hat{A} = I + \Delta A$ and $\hat{B} = \Delta B$. Then the discretised form of the state update can be written as:

$$
\begin{aligned}
h(t + \Delta t) &= \hat{A} h(t) + \hat{B} u(t) = (I + \Delta A)h(t) + \Delta B u(t) \\
&= h(t) + \Delta(Ah(t) + Bu(t)) = h(t) + \Delta h'(t).
\end{aligned} \tag{7}
$$

which implies it is a first order eular method. It is straightforward to show that $\tilde{A} = \hat{A}$ since $\tilde{A} = e^{A\Delta} = I + A\Delta + O(\Delta^2) = I + A\Delta = I + A\Delta = \hat{A}$. Similarly, we observe that $\tilde{B} = \bar{B} = \hat{B}$. Therefore, the discretisation used in the SSM method can be replaced with the zero-order hold method by substituting $\hat{A} = \tilde{A}$ and $\hat{B} = \tilde{B}$, we get:

$$
\begin{aligned}
h(t + \Delta t) &= \hat{A} h(t) + \hat{B} u(t) = \tilde{A} h(t) + \tilde{B} u(t) \\
&= (e^{\Delta A})h(t) + ((\Delta A)^{-1}(e^{\Delta A} - I) \cdot \Delta B)u(t) = (I + \Delta A + O(\Delta^2)h(t) + (\Delta B)u(t) \\
&= (I + \Delta A)h(t) + (\Delta B)u(t) = h(t) + \Delta Ah(t) + \Delta Bu(t) \\
&= h(t) + \Delta(Ah(t) + Bu(t)) = h(t) + \Delta h'(t).
\end{aligned} \tag{8}
$$

This shows that the zero-order hold discretisation method is equivalent to the Euler method, as both yield the same discrete update formula.

$\square$

Figure 1: (A) Illustration of Mamba Neural Operator. Input image patches are processed by following two distinct scanning paths (referred to as Bidirectional -Scan). Each sequence generated from these paths is passed through separate S6 blocks/ Cross S6 Blocks for independent processing. Afterwards, the outputs from the S6 blocks / Cross S6 Blocks are combined to form a feature map, resulting in the final output (Bidirectional-Merge). (B) and (C) are the detailed block of the S6 Block and Cross S6 Block respectively. The detail network architecture and definition of Cross S6 Block can be found in Appendix A.

**Why is Proposition 1 important for PDEs?** Proposition 1, which establishes the equivalence between the Zero-Order Hold (ZOH) method and the Euler method, is crucial for understanding Mamba's performance in solving partial differential equations (PDEs). This equivalence demonstrates that ZOH can be viewed as a more generalised and accurate variant of the Euler method. Specifically, while the Euler method is derived by applying the Taylor series expansion and truncating it at the first order, the ZOH method retains additional higher-order terms from the Taylor series, making it inherently more accurate. This distinction has significant implications when solving PDEs. Higher-order methods like ZOH provide better approximations of a system's behaviour without requiring excessively small step sizes $\Delta$, which are often necessary for the Euler method to achieve a similar level of accuracy. Smaller step sizes can result in increased computational cost and potential numerical instability. By utilising ZOH's higher-order accuracy, the Mamba architecture can handle a wide range of step sizes, ensuring both stability and convergence without compromising precision.

## 3.4 MAMBA FOR NEURAL OPERATORS

Neural Operators (Li et al., 2020b) aim to learn mappings between function spaces, providing a framework for solving partial differential equations (PDEs) and other problems involving continuous functions. It updates the value by an iterative method: $i_0 \rightarrow i_1 \rightarrow \ldots \rightarrow i_T$, where each $i_j$ (for $j = 0, 1, ..., T - 1$) maps to $\mathbb{R}^{d_v}$. Let the input be $a(x)$ and the output be $u(x)$. The input $a$, drawn from set $A$, is initially lifted to a higher-dimensional representation: $v_0(x) = P(a(x))$ where $P$ is a local transformation, typically parameterised by a fully-connected neural network. We then apply iterations to update $i_t \rightarrow i_{t+1}$ as defined in Definition 1. The final output: $u(x) = Q(v_T(x))$ is the result of projecting $v_T$ via the transformation: $Q : \mathbb{R}^{d_v} \rightarrow \mathbb{R}^{d_u}$. Each update from $i_t$ to $i_{t+1}$ involves the integration of a non-local integral operator $K$ and a local nonlinear activation function $\sigma$. One of the main results of this work is establishing the equivalence between neural operators and the Mamba framework. Therefore, we first introduce fundamental definitions stated in (Li et al., 2020b) that are essential for demonstrating this relationship.

**Definition 1.** *(Iterative updates): The update from $i_t \rightarrow i_{t+1}$ is defined as follows:*

$$i_{t+1}(x) := \sigma\left(W i_t(x) + K_\phi(a) i_t(x)\right), \quad \forall x \in D, \tag{9}$$

*where $K : A \times \Theta_K \to L(U(D; \mathbb{R}^{d_v}), U(D; \mathbb{R}^{d_v}))$ represents a mapping to bounded linear operators on $U(D; \mathbb{R}^{d_v})$, parameterised by $\phi \in \Theta_K$. The function $W : \mathbb{R}^{d_v} \to \mathbb{R}^{d_v}$ is a linear transformation, and $\sigma : \mathbb{R} \to \mathbb{R}$ is a nonlinear activation function applied component-wise.*

**Definition 2.** *(**Kernel integral operator** $K$): Define the kernel integral operator mapping in 1 by*

$$K_\phi(a)i_t(x) := \int_D \kappa_\phi(x, y, a(x), a(y))i_t(y)\, dy, \quad \forall x, \tag{10}$$

*where $\kappa_\phi : \mathbb{R}^{2(d+d_a)} \to \mathbb{R}^{d_v \times d_v}$ is a neural network parameterised by $\phi \in \Theta_K$.*

As mentioned in the previous section, we can be discrete SSM into the form of (5). This representation can be rewrite as (Liu et al., 2024): $\mathbf{h}_b = \mathbf{w}_T \odot \mathbf{h}_a + \sum_{i=1}^{T} \frac{\mathbf{w}_T}{\mathbf{w}_i} \odot (\mathbf{K}_i^\top \mathbf{V}_i)$. We define $\mathbf{V} = [\mathbf{V}_1; \ldots; \mathbf{V}T] \in \mathbb{R}^{T \times D_v}$, where $\mathbf{V}_i = u_{a+i-1}\Delta_{a+i-1} \in \mathbb{R}^{1 \times D_v}$, $\mathbf{K} = [\mathbf{K}_1; \ldots; \mathbf{K}_T] \in \mathbb{R}^{T \times D_k}$, where $\mathbf{K}_i = \mathbf{B}_{a+i-1} \in \mathbb{R}^{1 \times D_k}$ , and $\mathbf{Q} = [\mathbf{Q}_1; \ldots; \mathbf{Q}_T] \in \mathbb{R}^{T \times D_k}$, where $\mathbf{Q}_i = \mathbf{C}_{a+i-1} \in \mathbb{R}^{1 \times D_k}$. We further define $\mathbf{w} = [\mathbf{w}_1; \ldots; \mathbf{w}_T] \in \mathbb{R}^{T \times D_k \times D_v}$, where $\mathbf{w}_i = \prod_{j=1}^{i} e^{\mathbf{A}\Delta_{a-1+j}} \in \mathbb{R}^{D_k \times D_v}$, and $\mathbf{H} = [\mathbf{h}_a; \ldots; \mathbf{h}_b] \in \mathbb{R}^{T \times D_k \times D_v}$, where $\mathbf{h}_i \in \mathbb{R}^{D_k \times D_v}$. Finally, we set $\mathbf{Y} = [\mathbf{y}_a; \ldots; \mathbf{y}_b] \in \mathbb{R}^{T \times D_v}$, where $\mathbf{y}_i \in \mathbb{R}^{D_v}$ This formulation indicates that Gated Linear Attention (Yang et al., 2023) is actually a specific variant of Mamba. *We next present our main result is how neural operator layers share a comparable structural framework with time-varying SSMs, which, to the best of our knowledge, is established here for the first time.*

**Proposition 2.** *The hidden space in time-varying state-space models demonstrates a structural similarity to neural operator layers.*

*Proof.* We first rewrite the time-varying SSMs (5) as:

$$\mathbf{h}_b = \mathbf{w}_T \odot \mathbf{h}_a + \sum_{i=1}^{T} \frac{\mathbf{w}_T}{\mathbf{w}_i} \odot (\mathbf{K}_i^\top \mathbf{V}_i), \tag{11}$$

where $\mathbf{w}_T, \mathbf{w}_i, \mathbf{K}_i, \mathbf{V}_i$ are as defined previously.

To demonstrate that the hidden space update in our Mamba Operator has a similar structural framework to neural operator layers, we assume the shapes of $\mathbf{w}$ and $\mathbf{h}$ are $(T, D_k)$, represented as vectors. Our goal is to show that the iterative process in (5) aligns with that of Definition 1. Consider the first part of Definition 1, represented by $W_i(x)$. We set $W = \mathbf{W}_T$ and $i_t(x) = \mathbf{h}_a$, where $\mathbf{h}_a$ is the hidden state from the previous iteration. We then verify that $\mathbf{W}_T$ satisfies the properties of a linear transformation, ensuring consistency with the neural operator framework. We can proved this as follows: without loss of generality, let us assume that $W_T = W_1 = e^{\mathbf{A}\Delta_a}$. Next, we apply this transformation to a vector $x$ and check the conditions for linearity: $T(x + y) = T(x) + T(y)$ and $T(ax) = aT(x)$. By applying the Taylor expansion to $e^{\mathbf{A}\Delta_a}$ , then we get $I + A\Delta_a + O(\Delta_a^2)$. To show that $T(x + y) = T(x) + T(y)$, it suffices to demonstrate: $e^{A\Delta_a}(x_1 + x_1) = e^{A\Delta_a}(x_1) + e^{A\Delta_a}(x_2)$, we have:

$$
\begin{aligned}
e^{A\Delta_a}(x_1 + x_1) &= (I + A\Delta_a + O(\Delta_a^2)(x_1 + x_2) = I(x_1 + x_2) + A\Delta_a(x_1 + x_2) + O(\Delta_a^2)(x_1 + x_2) \\
&= Ix_1 + A\Delta_a(x_1) + O(\Delta_a^2)(x_1)Ix_2 + A\Delta_a(x_2) + O(\Delta_a^2)(x_2) \\
&= (I + A\Delta_a + O(\Delta_a^2))(x_1) + (I + A\Delta_a + O(\Delta_a^2))(x_2) \\
&= e^{A\Delta_a}(x_1) + e^{A\Delta_a}(x_2)
\end{aligned}
\tag{12}
$$

This shows $T(x + y) = T(x) + T(y)$. For the second condition, we want to show $T(\alpha x) = \alpha T(x)$, which is equivalent to demonstrating that $e^{A\Delta_a}(\alpha x) = \alpha e^{A\Delta_a}x$, we get:

$$
\begin{aligned}
e^{A\Delta_a}(\alpha x) &= (I + A\Delta_a + O(\Delta_a^2))(\alpha x) = I\alpha x + A\Delta_a\alpha x + O(\Delta_a^2)\alpha x \\
&= \alpha(Ix + A\Delta_a x + O(\Delta_a^2)x) = \alpha(e^{A\Delta_a}x)
\end{aligned}
\tag{13}
$$

Thus, we have shown that $e^{A\Delta_a}$ satisfies the two conditions, and hence it is a linear transformation. This shows the update in hidden space is the same as neural operator.

Secondly, we need to check the second part of Definition 1, which involves showing that:

$$K_\phi(a)i_t(x) := \sum_{i=1}^{T} \frac{\mathbf{w}_T}{\mathbf{w}_i} \odot (\mathbf{K}_i^\top \mathbf{V}_i) \tag{14}$$

has a similar structure.

According to Definition 2, it suffices to demonstrate that: $\int_D \kappa_\phi(x, y, a(x), a(y))i_t(y)\, dy = \sum_{i=1}^{T} \frac{\mathbf{w}_T}{\mathbf{w}_i} \odot (\mathbf{K}_i^\top \mathbf{V}_i)$. We assume the kernel $\kappa_\phi$ can be decomposed into a finite sum of separable basis functions: $\kappa_\phi(x, y, a(x), a(y)) = \sum_{i=1}^{T} \omega_i \varphi_i(x)\psi_i(y)$ such that $\omega_i$ is learnable weights for each basis function. and Basis functions capturing interactions between $x$ and $y$. Then we substitute it into the integral such that : $\int_D \sum_{i=1}^{T} \omega_i \varphi_i(x)\psi_i(y)v_t(y)\, dy = \sum_{j=1}^{T} \omega_i \int_D \varphi_i(x)\psi_i(y)v_t(y)\, dy$. We further discretise the domain $D$ into $T$ points $\{\mathbf{y}_i\}_{i=1}^{T}$ with corresponding weights $\Delta y$. The integral becomes $K_\phi(a)i_t(\mathbf{x}) \approx \sum_{i=1}^{T} \omega_i \sum_{j=1}^{T} \varphi_i(x)\psi_i(y_j)i_t(\mathbf{y}_j)\Delta y$. We represent $\varphi_i(\mathbf{x})$ as vector $\mathbf{K}_i$ and the input $i_t(\mathbf{y}_i)$ as vector $\mathbf{V}_i$: $\mathbf{K}_i = [\varphi_i(\mathbf{x}), \varphi_i(\mathbf{x}), \ldots, \varphi_i(\mathbf{x})]^\top \in \mathbb{R}^{1 \times D_k}$, $\mathbf{V}_i = [\psi_i(y_1)i_t(y_1)\Delta y, \ldots, \psi_i(y_T)i_t(y_T)\Delta y]^\top \in \mathbb{R}^{1 \times D_k}$. If we further factorise $\omega_i$ as $\frac{w_T}{w_i}$, where $w_T$ is a hyperparameter and $w_i$ represents a set of parameters to be learned, we obtain the update: $K_\phi(a)i_t(x) := \sum_{i=1}^{T} \frac{\mathbf{w}_T}{\mathbf{w}_i} \odot (\mathbf{K}_i^\top \mathbf{V}_i)$. Consequently, the neural operator layer shares a comparable structural framework with time-varying SSMs, demonstrating that the hidden space update in these models aligns with the iterative process in neural operator layers. $\qquad\square$

## 4 EXPERIMENTS AND DISCUSSION

In this section, we thoroughly describe the implementation setup and present experimental results to validate Mamba Neural Operators along with Transformers.

### 4.1 DATASET DESCRIPTION & IMPLEMENTATION PROTOCOL

⭐ **PDEs Selection.** We utilise datasets from PDEBench (Takamoto et al., 2022), a publicly available benchmark for partial differential equations (PDEs). We focus on three PDEs representing both stationary and time-dependent problems: Darcy Flow, Shallow Water 2D (SW2D), and Diffusion Reaction 2D (DR2D). All simulations are performed on a uniform grid. Detailed information about the datasets is provided in the Appendix.

⭐ **Implementation & Evaluation Protocol.** As Transformers have become the go-to architecture for PDE modelling and serve as the primary counterpart to SSM models, we selected three state-of-the-art Transformers as our baselines: GNOT (Hao et al., 2023), Galerkin Transformer (G.T.) (Cao, 2021), and OFormer (Li et al., 2022a). To achieve a fair comparison between Transformers and Mamba, we integrated the S6 block and Cross S6 block to replace self-attention and cross-attention in each model, creating modified versions of the original architectures. All three experimental methods initially adopt a linear attention mechanism as described in their original publications, while we evaluated two configurations for each of them: an implementation with standard softmax attention mechanism (w/S.A.) and a Mamba-enhanced implementation (our Mamba Neural Operator principle) (w/Mamba). All experiments were conducted on a single NVIDIA RTX 4090 GPU with 24GB of memory to ensure consistent and fair comparison conditions. Three metrics including Root Mean Squared Error (RMSE), Normalised RMSE (nRMSE), and Relative L2 Norm (RL2) were utilised for evaluation.

### 4.2 CHOSE YOUR WINNER: TRANSFORMER VS. MAMBA FOR PDES

We begin by evaluating the performance of Transformers, their variants, and Mamba on the Darcy Flow dataset, as presented in Table 1. The results demonstrate that incorporating Mamba consistently improves performance across all metrics and models. For GNOT, while the RMSE remains close, the nRMSE and RL2 values are reduced, indicating that Mamba effectively refines predictions. The G.T. sees the most significant enhancement, with the RMSE dropping by 40% when Mamba is used. This suggests that Mamba's design addresses the shortcomings of traditional Galerkin-type attention in capturing complex PDE dynamics. For OFormer, Mamba not only retains

Table 1: Quantitative comparison on Darcy Flow ($\beta = 1$) across three methods with linear attention (original version), softmax attention and Mamba. The performance is measured in terms of Root Mean Squared Error (RMSE), Normalised RMSE (nRMSE), and Relative L2 Norm (RL2), with the best-performing results highlighted.

| METHOD | TYPE | DARCYFLOW | | |
|---|---|---|---|---|
| | | RMSE↓ | nRMSE↓ | RL2↓ |
| GNOT (Hao et al., 2023) | Galerkin | 0.0070 | 0.0485 | 0.0370 |
| w/S.A. | Softmax | 0.0061 | 0.0394 | 0.0299 |
| w/Mamba (MNO) | Mamba | 0.0061 | 0.0367 | 0.0297 |
| G.T. (Cao, 2021) | Galerkin | 0.0188 | 0.2027 | 0.1261 |
| w/S.A. | Softmax | 0.0103 | 0.1050 | 0.0648 |
| w/Mamba (MNO) | Mamba | 0.0061 | 0.0382 | 0.0286 |
| OFormer (Li et al., 2022a) | Normalised | 0.0054 | 0.0253 | 0.0242 |
| w/S.A. | Softmax | 0.0066 | 0.0324 | 0.0323 |
| w/Mamba (MNO) | Mamba | 0.0054 | 0.0244 | 0.0241 |

Table 2: Quantitative comparisons on Shallow Water 2D (SW2D) and Diffusion Reaction 2D (DR2D) across three methods with linear attention (original version) and Mamba. The performance is measured in terms of Root Mean Squared Error (RMSE), Normalised RMSE (nRMSE), and Relative L2 Norm (RL2), with the best-performing results highlighted in green.

| METHOD | TYPE | SW2D | | | DR2D | | |
|---|---|---|---|---|---|---|---|
| | | RMSE↓ | nRMSE↓ | RL2↓ | RMSE↓ | nRMSE↓ | RL2↓ |
| GNOT (Hao et al., 2023) | Galerkin | 0.0026 | 0.0025 | 0.0027 | 0.0567 | 0.6953 | 0.7233 |
| w/Mamba (MNO) | Mamba | 0.0023 | 0.0022 | 0.0024 | 0.0060 | 0.0811 | 0.0570 |
| G.T. (Cao, 2021) | Galerkin | 0.0037 | 0.0035 | 0.0038 | 0.0083 | 0.1259 | 0.0723 |
| w/Mamba (MNO) | Mamba | 0.0013 | 0.0013 | 0.0014 | 0.0012 | 0.0183 | 0.0099 |
| OFormer (Li et al., 2022a) | Normalised | 0.0020 | 0.0020 | 0.0021 | 0.0177 | 0.2681 | 0.1559 |
| w/Mamba (MNO) | Mamba | 0.0021 | 0.0021 | 0.0022 | 0.0123 | 0.1712 | 0.1134 |

the strong baseline performance but also achieves improvements across all metrics. The reduction in RL2 indicates that Mamba's mechanism is better at mapping the solution space of PDEs with higher precision. Mamba also demonstrates an enhanced ability to capture the complex spatial correlations inherent to Darcy Flow more effectively.

On the SW2D dataset, Mamba consistently outperforms the original Transformer models across all metrics. GNOT with Mamba achieves a lower RMSE and RL2, demonstrating Mamba's ability to capture complex flow dynamics. The G.T. shows the most significant improvement, with RMSE of 65% reduction—highlighting Mamba's superior capability in accurately representing the system's behaviour. For OFormer, Mamba maintains comparable values but increases the RL2. On the DR2D dataset, the Mamba-enhanced models exhibit even more substantial gains. The G.T. sees a dramatic reduction in RMSE and RL2, showing Mamba's strength in handling the complex dynamics.

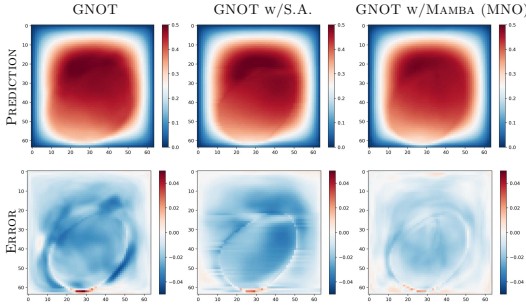

Figure 2: Results of prediction map and error map of the GNOT across three versions: Galerkin attention, Softmax attention, and Mamba.

The results across all datasets demonstrate a clear advantage of the Mamba Neural Operator over Transformer architectures for PDEs. While Transformers are effective at capturing dependencies and patterns, Mamba's specialised attention mechanisms provide a more understanding of the complex dynamics involved. By leveraging its unique cross-attention and self-attention blocks, Mamba not only achieves lower error rates but also enhances the stability and precision of predictions, particularly in highly nonlinear systems. *These results suggest that Mamba enhances the expressive power and accuracy of neural operators, indicating that it is not just a complement to Transformers*

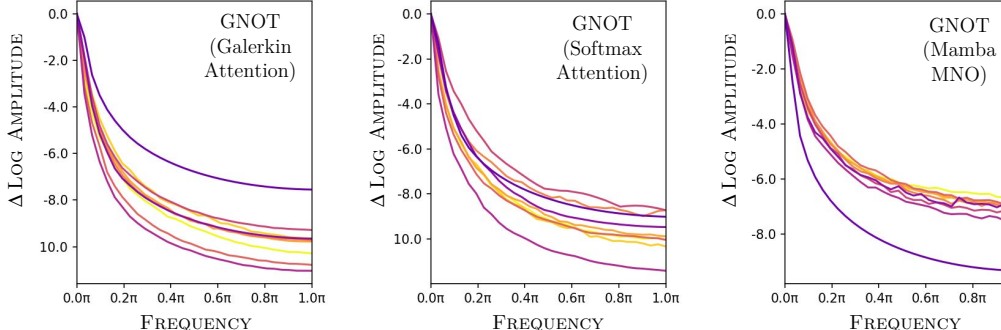

Figure 3: Fourier analysis comparing three GNOT versions: Galerkin attention, Softmax attention, and Mamba. The $\Delta$ log amplitude shows how each model handles frequency components. We calculate the change by comparing the log amplitude at the center $(0.0\ \pi)$ and boundary frequencies $(1.0\ \pi)$. For clarity, only half-diagonal components of the 2D Fourier-transformed feature maps are shown.

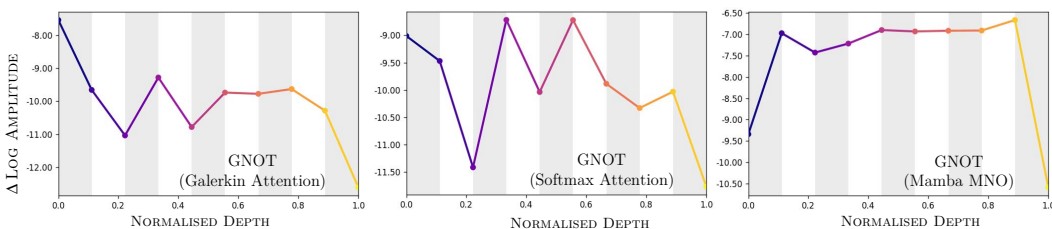

Figure 4: $\Delta$ log amplitudes for Galerkin attention, Softmax attention, and Mamba. Gray regions indicate the operator, and white regions show MLP. Mamba shows a more stable response across frequencies.

*but a superior framework for PDE-related tasks*, bridging the gap between efficient representation and accurate solution approximation.

We further validate Mamba's potential through visualisations, as shown in Figure 2. The prediction and error maps reveal that Mamba consistently outperforms all Transformer variants, delivering more accurate solutions with lower error across challenging regions. Mamba handles fine details, particularly in capturing sharp gradients and subtle variations that standard attention mechanisms often miss. Compared to the Galerkin and Softmax attention Transformer models, Mamba reduces error propagation and improves spatial coherence. More Visualization result can be seen in Appendix C.

### 4.3 WHY THE WINNER WINS: BREAKING DOWN MAMBA'S WIN

We aim to explore why Mamba outperforms Transformers by examining the frequency response of feature maps. This analysis helps us understand how each model handles high-frequency signals and evaluate its ability to maintain stability and robustness. The results in Figure 3 compare the frequency response of three GNOT variants: Galerkin attention, Softmax attention, and Mamba. The Galerkin version shows a sharp decline in high-frequency components, indicating underfitting and loss of fine details. The Softmax version retains more high frequencies but risks instability and noise sensitivity. Mamba, on the other hand, demonstrates a balanced suppression of high-frequency signals, maintaining stability and robustness. The change in log amplitude across the frequency range is more uniform, indicating that Mamba effectively balances between capturing necessary high-frequency information and filtering out noise. This controlled response across the spectrum highlights why Mamba is better suited for PDEs.

Figure 4 shows the $\Delta$ log magnitudes across the normalised depth for the Galerkin attention, Softmax attention, and Mamba versions of GNOT. The Galerkin and Softmax versions exhibit sharp

Table 3: Comparisons with different query positions using nRMSE.

| METHOD | Query Positions | |
|---|---|---|
| | Identical | Diagonal |
| OFormer | 0.0253 | 0.0318 |
| w/S.A. | 0.0324 | 0.0382 |
| w/Mamba | 0.0244 | 0.0314 |

Table 4: Comparisons with different dataset sizes using nRMSE.

| METHOD | Dataset Sizes | | | |
|---|---|---|---|---|
| | 9K | 5K | 2K | 1K |
| GNOT | 0.0485 | 0.0567 | 0.0777 | 0.1174 |
| w/S.A. | 0.0394 | 0.0400 | 0.0526 | 0.0776 |
| w/Mamba | 0.0367 | 0.0376 | 0.0481 | 0.0617 |

fluctuations, indicating instability and inconsistent feature extraction at different depths. In contrast, Mamba maintains a steady and flat profile, reflecting robust and stable feature extraction. The gray and white bands indicate the alternating roles of the operator and NLP components, further emphasising Mamba's balanced performance across layers, making it ideal for handling complex PDEs.

### 4.4 ABLATION STUDY: FINAL BATTLES, WINNER TAKES ALL

**Mamba vs. Transformers in Misalignment: A Battle for Query Positioning.** Table 3 compares nRMSE performance across two query scenarios: Identical positions (input and query points are the same) and Diagonal positions (shifted inputs creating a mismatch). Experiments are done using Darcy Flow. While prior work shows that Transformers handle inconsistent input-query positions well (Li et al., 2022a), our results demonstrate a clear advantage of Mamba in both configurations. For Identical query positions, Mamba version achieves the lowest error, outperforming OFormer and its softmax variant, demonstrating *Mamba's superior ability to capture relationships when input and query points are perfectly aligned.* For Diagonal query positions, where inputs and queries are misaligned, Mamba achieves the best performance compared to OFormer and its variant, demonstrating its superior *ability to generalise under spatial shifts.*

**Scaling Down Without Sacrifice: A Battle for Resilience with Limited Data.** Table 4 compares nRMSE performance across different dataset sizes for GNOT, GNOT with softmax attention (w/ S.A.), and GNOT with Mamba. Experiments are carried out using Darcy Flow. As dataset size decreases, Mamba consistently achieves the lowest error, demonstrating superior performance and robustness in data-scarce scenarios. For instance, with the smallest dataset (1K), Mamba achieves an nRMSE of 0.0617, significantly lower than GNOT's and GNOT w/ S.A.'s, showcasing its resilience and generalisation capability even with limited data. *This highlights Mamba's efficiency in learning meaningful representations with fewer data points, making it a powerful choice for real-world applications where data availability is a constraint.*

## 5 CONCLUSION

We have introduced the concept of the Mamba Neural Operator (MNO), a framework that redefines how neural operators approach PDEs by integrating structured state-space models. Unlike closely related works, we formalise this connection by providing a theoretical understanding that demonstrates how neural operator layers share a comparable structural framework with time-varying SSMs, offering a fresh perspective on their underlying principles. Experimental results show that MNO significantly enhances the expressive power and accuracy of neural operators across various architectures and PDEs. This indicates that MNO is not merely a complement to Transformers, but a superior framework for PDE-related tasks, bridging the gap between efficient representation and precise solution approximation.

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
