# Supplmentary Material
# Mamba Neural Operator: Who Wins? Transformers vs. State-Space Models for PDEs

In this supplementary material, we provide additional details regarding our methodology and a more comprehensive description of the dataset used in our experiments.

## A    Supplementary Information

### A.1    Preliminaries: Transformer.

In each Transformer layer, an attention mechanism enables interaction between inputs at varying positions, followed by a position-wise fully connected network applied independently to each position. Specifically, the attention mechanism involves projecting an intermediate representation into three components—query $Q \in \mathbb{R}^{N \times d_k}$, key $K \in \mathbb{R}^{N \times d_k}$, and value $V \in \mathbb{R}^{N \times d_v}$—using three separate position-wise linear layers. These representations are then used to calculate the output as:

$$\text{Attention}(Q, K, V) = \text{softmax}\left(\frac{QK^T}{\sqrt{d_k}}\right)V, \tag{S.1}$$

of which the memory complexity is $O(n^2)$. To reduce the computational inefficiency, Galerkin-type attention was proposed by (Cao, 2021) to remove Softmax attention with linear complexity. It defines as follows:

$$\text{Attention}_g(Q, K, V) = \frac{Q(\widetilde{K}^T \widetilde{V})}{d}, \tag{S.2}$$

in which $\widetilde{\phantom{x}}$ denotes layer normalisation, as described in (Lei Ba et al., 2016). The Galerkin-type attention mechanism involves two matrix product operations, resulting in a computational complexity of $O(nd^2)$. This reduces the sequence length dependency to only $O(n)$.

### A.2    Network Architecture.

As depicted in Figure 1 from the main paper, the data processing pipeline in our Mamba Neural Operator (MNO) is composed of three key stages: Bi-Directional Scan Expand, S6/Cross S6 Block, and Bi-Directional Scan Merge. When solving PDEs over a fixed grid, the input data can be structured as grid-based data, similar to an image. In the first stage, Bi-Directional Scan Expand, the MNO unfolds the input data into sequences by traversing the grid along two distinct paths. These sequences, representing input patches, are processed independently in the next step. The second stage, S6/Cross S6 Block, involves processing each patch sequence using either an S6 or Cross S6 block, depending on the model variation being employed. For instance, in the enhanced version of Mamba, the GNOT model utilises a Cross S6 block followed by an S6 block for further refinement. Finally, in the Bi-Directional Scan Merge stage, the processed sequences are reshaped and merged back together to generate the output map, completing the data forwarding process. This structured approach allows the MNO to efficiently handle grid-based input data, enabling scalable solutions for PDEs.

### A.3    Definition of Cross S6.

Let $x$ and $x'$ be two independent input vectors. Each input is processed through two independent linear transformation, resulting in corresponding parameter sets $(B, C, \Delta)$ for $x$ and $(B', C', \Delta')$ for $x'$. Specifically, these transformations are defined as:

$$B, C, \Delta = \text{Linear}_x(x),$$
$$B', C', \Delta' = \text{Linear}_{x'}(x'), \tag{S.3}$$

where $\text{Linear}_x$ and $\text{Linear}_{x'}$ are the respective linear layers applied to $x$ and $x'$.

Next, the parameters $(\tilde{B}, \tilde{C}, \tilde{\Delta})$ are computed by combining the updated values from both inputs according to the following equations:

$$\tilde{B} = B + qB',$$
$$\tilde{C} = C + qC', \tag{S.4}$$
$$\tilde{\Delta} = \Delta + q\Delta',$$

where $q$ is a scalar ratio controlling the contribution of the second input $x'$ to the combined output. Once we have these updated parameters, we apply the State Space Model (SSM) to compute the final output y.

## B FURTHER IMPLEMENTATION DETAILS

**Darcy Flow.** The two-dimensional Darcy Flow equation defines as follows:

$$\begin{cases} -\nabla \cdot (a(x,y)\nabla u(x,y)) = f(x,y), & \text{for } (x,y) \in \Omega, \\ u(x,y) = 0, & \text{for } (x,y) \in \partial\Omega, \end{cases} \tag{S.5}$$

where $a(x,y)$ is the diffusion coefficient, $u(x,y)$ is the solution respectively, and $\Omega = (0,1)^2$ is a square domain. In Darcy Flow, the force term $f(x,y)$ is set to be a hyperparameter $\beta$, which influences the scale of the solution $u(x,y)$. Experiments were performed on the steady-state solution of the 2D Darcy Flow over a uniform square domain. The goal is to approximate the solution operator $S$ defined by:

$$S : a \mapsto u, \quad \text{for } (x,y) \in \Omega, \tag{S.6}$$

with $a(x,y)$ and $u(x,y)$ as previously defined. Similar as PDEBench (Takamoto et al., 2022) protocol, we used only $\beta = 1.0$ and we divided the training and testing ratio into 9:1 which contains 9,000 samples for training and 1,000 samples for testing.

**Shallow Water.** We conducted experiments on the two-dimensional Shallow Water equations, which are effective for modeling free-surface flow problems. The equations are formulated as follows:

$$\partial_t h + \partial_x(hu) + \partial_y(hv) = 0,$$
$$\partial_t(hu) + \partial_x\left(u^2 h + \tfrac{1}{2}g_r h^2\right) = -g_r h\, \partial_x b, \tag{S.7}$$
$$\partial_t(hv) + \partial_y\left(v^2 h + \tfrac{1}{2}g_r h^2\right) = -g_r h\, \partial_y b,$$

where $u = u(x,y,t)$ and $v = v(x,y,t)$ represent the velocities in the horizontal and vertical directions, respectively, and $h = h(x,y,t)$ denotes the water depth. The term $b = b(x,y)$ stands for the spatially varying bathymetry, and $g_r$ is the gravitational acceleration.

The dataset simulates a 2D radial dam-break scenario within a square domain $\Omega = [-2.5, 2.5]^2$ over the time interval $t \in [0,1]$. The initial condition is defined by:

$$h(t=0, x, y) = \begin{cases} 2.0, & \text{if } \sqrt{x^2 + y^2} < r, \\ 1.0, & \text{if } \sqrt{x^2 + y^2} \geq r, \end{cases} \tag{S.8}$$

where the radius $r$ is randomly drawn from a distribution $D(0.3, 0.7)$.

Our objective is to approximate the solution operator $S$, defined as:

$$S : h|_{t \in [0,t']} \mapsto h|_{t \in (t',T]}, \quad (x,y) \in \Omega, \tag{S.9}$$

with $t' = 0.009$ s and $T = 1.000$ s. Here, $h = h(x,y,t)$ represents the water depth over time.

Each sample in the dataset is discretized on a spatial grid of $128^2$ points and a temporal grid of 101 time steps. The first 10 time steps are used as input to the model, while the remaining 91 time steps serve as the target output. Following the protocol established by PDEBench (Takamoto et al., 2022), the dataset consists of 900 samples for training and 100 samples for testing.

**Diffusion Reaction.** The Diffusion Reaction equations are expressed as:

$$\begin{aligned}
\partial_t u &= D_u \partial_{xx} u + D_u \partial_{yy} u + R_u, \\
\partial_t v &= D_v \partial_{xx} v + D_v \partial_{yy} v + R_v,
\end{aligned} \tag{S.10}$$

where the activator and inhibitor are represented by the functions $u = u(x,y,t)$ and $v = v(x,y,t)$. In addition, these two variables are non-linearly coupled variables. These functions describe the interaction between the activator and inhibitor in the system. $D_u = 1 \times 10^{-3}$ and $D_v = 5 \times 10^{-3}$ are the diffusion coefficients for the activator and inhibitor, respectively.

The reaction terms for the activator and inhibitor are then defined as follows:

$$R_u(u,v) = u - u^3 - k - v, \quad R_v(u,v) = u - v, \tag{S.11}$$

with $k = 5 \times 10^{-3}$. The simulation is performed over the domain $\Omega = [-1,1]^2$ with the time interval $t \in [0,5]$. The solution operator $S$ is defined as:

$$S : \{u,v\}_{t \in [0,t']} \mapsto \{u,v\}_{t \in (t',T]}, \quad (x,y) \in \Omega, \tag{S.12}$$

where $t' = 0.045$ s and $T = 5.000$ s, and the spatial domain is $\Omega = [-1,1]^2$. Here, $u = u(x,y,t)$ and $v = v(x,y,t)$ represent the activator and inhibitor, respectively. In this dataset, we follow the same discretization scheme similar to the Shallow Water equation, where each sample is downsampled to a spatial resolution of $128^2$ and a temporal resolution of 101 time steps (with 10 for input and rest of the 91 for target). Similar as the PDEBench protocol (Takamoto et al., 2022), the dataset includes 900 samples for training and 100 samples for testing.

## C  EXTENDED VISUAL RESULTS

This section extends the visual results presented in the main paper, providing a deeper comparison of the predictive performance and error distribution across different models. The supplementary figures illustrate the impact of incorporating Mamba on prediction accuracy and spatial coherence, further validating its advantages over traditional Transformer-based approaches.

Figure S.1 presents the prediction and error maps for the Galerkin Transformer (G.T.) and OFormer across three configurations: Galerkin attention, standard softmax attention, and Mamba (MNO). The results show that Mamba consistently achieves lower prediction errors, especially in regions with high variability, highlighting its ability to capture complex dynamics with greater precision compared to other configurations.

Figure S.2 and Figure S.3 provide visualised predictions over time for the Shallow Water and Diffusion Reaction datasets, respectively, using the original Galerkin Transformer and its Mamba-enhanced version. For the Shallow Water dataset, the Mamba-integrated model better preserves fine details and the circular wavefronts as time progresses, reflecting its superior capability to maintain spatial coherence. Similarly, in the Diffusion Reaction dataset, Mamba reduces the spread of error and better approximates the reference solution, demonstrating improved stability and generalisation in long-term simulations.

Overall, these visualisations clearly indicate that incorporating Mamba significantly enhances predictive accuracy and robustness, making it a superior choice for capturing intricate spatial-temporal patterns in complex PDE systems.

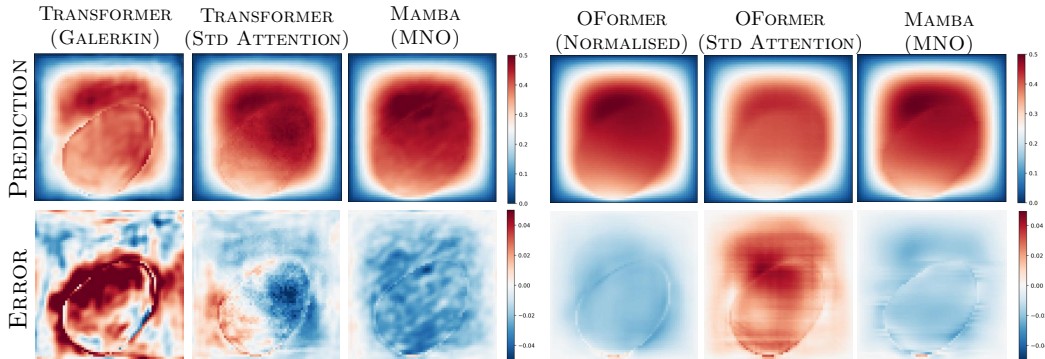

Figure S.1:   Results of prediction map and error map of the Galerkin Transformer and OFormer across three versions: Galerkin attention, Softmax attention, and Mamba.

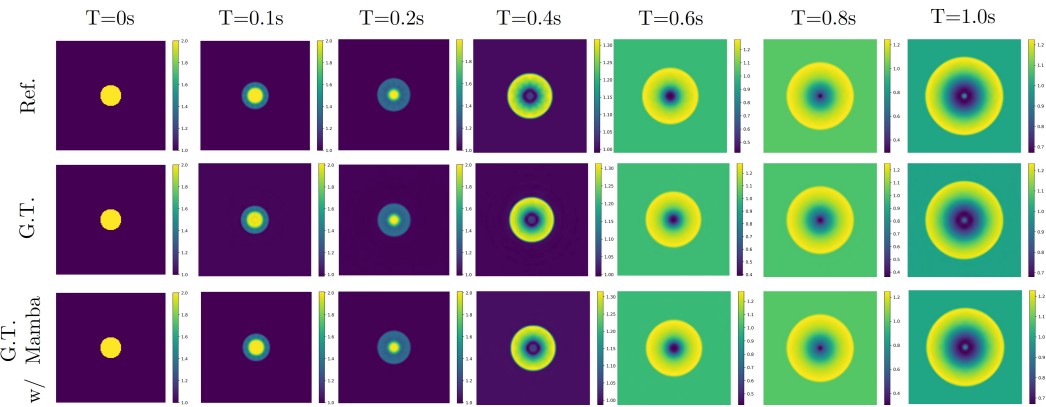

Figure S.2:   Visualised prediction on Shallow Water dataset using Galerkin Transformer (G.T.) across the original and Mamba version.

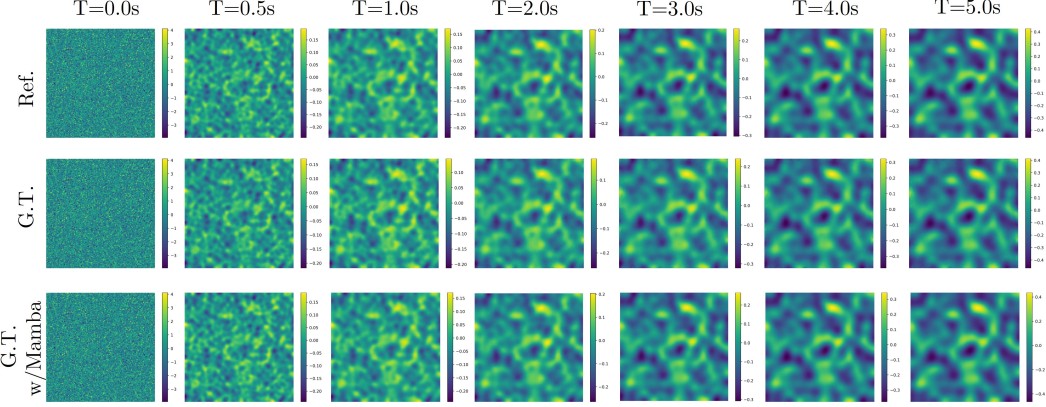

Figure S.3:  Visualised prediction on Diffusion Reaction dataset using Galerkin Transformer (G.T.) across the original and Mamba version.