# OpenReview forum: "Mamba Neural Operator: Who Wins? Transformers vs. State-Space Models for PDEs"
_ICLR.cc/2025/Conference — ICLR 2025 Conference Withdrawn Submission_

### Official Review · Reviewer_vEWw · 2024-10-18

**Soundness:** 2
**Presentation:** 2
**Contribution:** 1
**Rating:** 3
**Confidence:** 4

**Summary:**

This paper introduces Mamba in the PDE-solving area. It establishes a formal theoretical connection between structured state-space models (SSMs) and neural operators. The proposed method performs relatively well on the considered datasets.

**Strengths:**

1. The summary of related work is detailed.
2. The paper establishes a theoretical connection between the Mamba architecture and neural operators.
3. Although not comprehensive, the paper includes experiments on several datasets and compares the proposed method with some baseline models.

**Weaknesses:**

1. **Presentation Issues:** The paper lacks line numbers, which are essential for precise referencing during reviews. Additionally, the appendix is included in the supplementary materials rather than in the main document, which hampers the ease of accessing critical information.
2. **Writing Clarity:** The manuscript’s writing requires enhancement, as some sentences lack logical coherence. For instance, the statement: *“While Transformers dominate in areas like foundational models and computer vision, the application of SSMs, particularly the Mamba architecture, to neural operators for PDEs is still underexplored.”* The connection between Transformers’ dominance in certain fields and the underexplored application of SSMs in PDEs is unclear.
3. **Figure 1 Interpretation:** The purpose and contribution of Figure 1 are not clear. The main paper contains excessive mathematical details that could be better placed in the appendix. And a detailed description of the main architecture of MNO is necessary. Specifically, does MNO include a patchification layer as shown in Figure 1? Furthermore, since your baseline models operate on mesh points directly without patchifying, how does this influence your comparisons?
4. **Novelty Concern:** The paper’s novelty appears limited. Introducing the Mamba architecture into PDE solving is not new. The approach seems to involve simply replacing the Transformer block with a Mamba block without significant modifications. It would be helpful to explain why PDE-specific adaptations are not needed. Additionally,  while you theoretically demonstrate that the MNO is a type of neural operator, this does not enhance the understanding of the benefits of employing Mamba in PDE solving.
5. **Efficiency Evaluation:** It is well-known that Mamba offers efficiency advantages over Transformers in long sequence data. However, your experiments do not include efficiency comparisons with baseline models, which is a missed opportunity to highlight Mamba’s benefits.
6. **Dataset and Baseline Limitations:** The datasets and baselines used are not comprehensive enough. The datasets selected are relatively simple. Incorporating more complex datasets, such as the 2D Navier-Stokes dataset from FNO [1], would strengthen your evaluation. Additionally, there are other Transformer-based solvers beyond those utilizing linear attention, such as Transolver [2], which computes attention based on learned physical states.
7. **Implementation Details:** The implementation details provided are insufficient. Did you keep all hyperparameters consistent across the three variants? What is the total parameter count for each model? It is hard to believe that merely replacing the Transformer block with a Mamba block results in significant performance improvements, especially since such gains have not been observed in NLP and CV domains. Providing more detailed implementation information or sharing inference code would greatly enhance the paper’s credibility.
8. **Clarity of Figure 3:** The representation in Figure 3 is not clear. What do the different colored lines mean, and do these lines correspond to any ground truth? The sentence *“The change in log amplitude across the frequency range is more uniform, indicating that Mamba effectively balances between capturing necessary high-frequency information and filtering out noise.”* is difficult to understand. It would be helpful if you could provide more explanations or clarify this statement.
9. **Irregular data:** The paper focuses solely on grid data. I wonder what distinctions do you see between grid-based PDE data and images. Futhermore, how would you extend the MNO to handle irregular data, considering that linear Transformers can readily address such types?

[1] Fourier Neural Operator for Parametric Partial Differential Equations

[2] Transolver: A Fast Transformer Solver for PDEs on General Geometries

**Questions:**

See Weaknesses.

---

### Official Review · Reviewer_TBE1 · 2024-10-30

**Soundness:** 1
**Presentation:** 3
**Contribution:** 1
**Rating:** 3
**Confidence:** 5

**Summary:**

This paper integrates Mamba and designs the MNO model to solve PDE equations. It compares Transformer-based models and draws some inconsequential conclusions.

**Strengths:**

Mathematical derivations are complete, coherent, and consistent. Combines the mathematical frameworks of Mamba (Gu, 2023) and Neural Operator (Kovachki, 2023).

**Weaknesses:**

(1) This paper sets the Transformer as a target for comparison in studying neural operators, disregarding all other neural operators. However, the Transformer is neither mainstream nor representative in the neural operator field (see Questions part for details). This setting limits the value of the conclusions drawn in the paper.

(2) The paper uses RMSE as a metric for evaluating PDE solutions but fails to address critical properties of PDE solving. This undermines the persuasiveness of the proposed model, suggesting that MNO may be a better tool for approximation series rather than an effective PDE solver. The paper overlooks the generalization capability of Neural Operators in Banach spaces—an essential feature of neural operators. It also fails to discuss resolution invariance or any specific physical or mathematical consistency with respect to the target PDE system. This makes it difficult to determine if the model is locally fitting a few PDE solutions or genuinely learning the target operator.

In summary, the significance and value of this paper are unclear, resulting in limited research impact.

**Questions:**

Personally, I find some problems with the description of Transformer-based PDE solving in the introduction part. First, Transformer architectures are not mainstream in Neural Operators, so many models such as GNO, FNO, KNO, CNO, and DeepONet, which tend to prefer more lightweight structures instead of Transformer. This raises questions about the framing in the introduction. The authors mention, "While Transformers are popular for PDE modelling, they have limitations in handling continuous data and high-resolution grids." However, in PDE problems, Transformer architectures often perform poorly at lower resolutions (e.g., 64x64 or 128x128) and achieve better results at higher resolutions (such as 720x1440), as evidenced by AFNO and FourCastNet.

In summary, there are two questions.
1. Please explain why Transformers were chosen as the comparison baseline while other operator methods were overlooked.
2. Please clarify the reasoning behind the claim that Transformers perform poorly at high resolutions?

---

### Official Review · Reviewer_cEgC · 2024-11-01

**Soundness:** 3
**Presentation:** 1
**Contribution:** 2
**Rating:** 3
**Confidence:** 5

**Summary:**

Introduces Mamba Neural Operator as an alternative to transformer.

**Strengths:**

1) Theoretical Connection between Mamba and Neural Operator.

2) MNO addresses long-range dependencies and continuous dynamics better than a transformer.

**Weaknesses:**

1) The presentation could be improved by addressing notational inconsistencies.

2) Lack of Novelty as compared to Vision Mamba.

3) Weak baselines. SOTA transformers such as Transolver, Latent Neural Operator, DPOT, LSM, etc.

4) Missing model parameters, training time, inference time comparison and MNO Hyperparamters.

**Questions:**

1) MNO uses bidirectional scans, which are from two directions. As continuity is essential for PDEs, are there other directions you can also consider, like ZigZag Mamba? How is it different from vision mamba?

2) I don't get why we need to consider the cross-scan in MNO. Have you done ablation removing the cross scan, as you have already considered the bidirectional scan? Or does it help boost performance? Have you tried running naive vision Mamba as a baseline?

3) I found the paper difficult to read on Page 6 after the Kernel Integral Operator definition and spent quite a lot of time trying to understand it better. Notation could be more precise. Presentation could be improved as in the main script, MNO architecture can be adequately described. Use consisent notation as in Equ 5 (Captial U and small u) and in proposition 2.

4) How is Proposition 2 Kernel Integral when making specific points out of domain? Isn't it more like an approximation? Also, part 2 proof is confusing.

5) Could you please provide a more comprehensive description of the MNO architecture? If you are adding nonlinearity such as FNO, hyperparameters, etc. How did you finetune the hyperparamters?

6) Does MNO hold a universal approximation of an operator like FNO?

7) See Weakness

---

### Official Review · Reviewer_G8Nk · 2024-11-01

**Soundness:** 2
**Presentation:** 2
**Contribution:** 2
**Rating:** 3
**Confidence:** 4

**Summary:**

This paper presents the Mamba Neural Operator (MNO), a new approach designed to address the challenges in modeling and solving partial differential equations (PDEs) using neural networks. The MNO integrates structured state-space models (SSMs) into neural operators, allowing it to capture long-range dependencies and continuous dynamics more effectively than existing Transformer-based methods.

**Strengths:**

1. The topic explored in this paper is interesting and engaging.

2. The paper offers some theoretical insights that enhance our understanding of operator learning.

3. The experiments conducted are thorough.

**Weaknesses:**

1.  The paper feels somewhat rushed, which has resulted in several typos that hinder understanding of the content. Here are some specific issues I've noticed:

   a. In Section 3.1, there is a mistake in one of the formulas ($n=1^N$), which could confuse readers trying to grasp the main concepts.

   b. In Section 3.2, it’s unclear how $B$, which is an n-dimensional complex vector, and $u(t)$, which is an L-dimensional real vector, are multiplied in Equation 3. This lack of clarity raises questions about the mathematical operations being performed.

   c. In Equation 4, the variables $A$ and $B$ are introduced, but it's unclear whether they are the same as what’s defined in Equation 3.

   d. In Equation 5, there seems to be inconsistency with the notation: are $U_a$ and $u_a$ meant to represent the same entity? Additionally, is $\Delta_a$ the same as $\Delta a$?

   e. In Equation 6, the term "Drop" is used without adequate explanation, leaving readers uncertain about its intended meaning.

2. In Section 3.3 (Page 5), the authors should specify which order of terms they are referring to when they mention "higher-order terms." Providing this detail is essential for understanding the implications of their claims. Furthermore, if the authors intend to use this aspect of the ZOH to demonstrate the superiority of Mamba, they need to establish that traditional transformers cannot align with the Taylor expansion.

3. In the experimental results, the authors should report the differences in training and inference times between Mamba and other methods, particularly focusing on inference time, as this is crucial for neural operators. Additionally, I recommend that the authors include comparisons with traditional numerical methods.

4. The authors primarily focus their experiments on a two-dimensional setting. However, it raises an important question: when extending to a three-dimensional setting, which approach would be more suitable—Mamba or transformers? It’s worth mentioning that there are already existing works that utilize transformers for 3D neural operators [1].

[1] Li Z, Liu T, Peng W, et al. A transformer-based neural operator for large-eddy simulation of turbulence[J]. arXiv preprint arXiv:2403.16026, 2024.

**Questions:**

Please see the weaknesses part.

---

### Note · Authors · 2024-11-14

I have read and agree with the venue's withdrawal policy on behalf of myself and my co-authors.